# AE-RTISNet: Aeronautics Engine Radiographic Testing Inspection System Net with an Improved Fast Region-Based Convolutional Neural Network Framework

**Zhi-Hao Chen *** and Jyh-Ching Juang

Department of Electrical Engineering, National Cheng Kung University, Tainan 701, Taiwan;
8202019@gs.ncku.edu.tw
* Correspondence: n28061072@gs.ncku.edu.tw; Tel.: +886-6-2757575 (ext. 62400)

**Abstract:** To ensure safety in aircraft flying, we aimed to use deep learning methods of nondestructive examination with multiple defect detection paradigms for X-ray image detection. The use of the fast region-based convolutional neural network (Fast R-CNN)-driven model was to augment and improve the existing automated non-destructive testing (NDT) diagnosis. Within the context of X-ray screening, limited numbers and insufficient types of X-ray aeronautics engine defect data samples can, thus, pose another problem in the performance accuracy of training models tackling multiple detections. To overcome this issue, we employed a deep learning paradigm of transfer learning tackling both single and multiple detection. Overall, the achieved results obtained more than 90% accuracy based on the aeronautics engine radiographic testing inspection system net (AE-RTISNet) retrained with eight types of defect detection. Caffe structure software was used to perform network tracking detection over multiple Fast R-CNNs. We determined that the AE-RTISNet provided the best results compared with the more traditional multiple Fast R-CNN approaches, which were simple to translate to C++ code and installed in the Jetson™ TX2 embedded computer. With the use of the lightning memory-mapped database (LMDB) format, all input images were 640 × 480 pixels. The results achieved a 0.9 mean average precision (mAP) on eight types of material defect classifier problems and required approximately 100 microseconds.

**Keywords:** fast R-CNN; R-CNN; NDT; X-ray; transfer learning

## 1. Introduction

In the National Transportation Safety Board (NTSB) global aviation safety notice [1], it is stated that "An abnormal engine could cause a serious aircraft accident". To mitigate the risk of aircraft accidents, it is thus of paramount importance to perform quality-assured engine inspections, manufacturing, and overhauls. Indeed, if the inspection and maintenance tasks of airplane engines are negligent, flight safety problems may result.

This was evidenced in an event concerning a Boeing 737-700 passenger aircraft on April 2018. In this event, a serious accident occurred when the engine blade ruptured and pierced the cabin of the aircraft, damaging the fuselage and causing injury to the passengers. The news report shown in Figure 1 depicts a failed fan blade as well as a fracture surface with fatigue. The main cause of the accident was that engine blades inspection was not implemented, and the hidden fatigue defects in the engine blades were not detected by the inspector. However, defects can occur during material welding maintenance work, including micro-cracks, incomplete fusion, voids, porosity, blowholes, and inclusion spatter [2,3]. Once engine blades with hidden cracks have exceeded their service life, fatigue

micro-cracks on metal surfaces causes the cracks to continuously expand. The engine aeronautics composite materials (ACM) [4] are mainly made of aluminum (titanium) alloy composite materials and related spare parts.

According to the Federal Aviation Administration (FAA) maintenance review board (MRB) [5], each online aerial engine should pass the period A, B, C, and D level checks for work, including maintenance, repair, and overhaul (MRO) [6]. These sites will be checked with non-destructive testing (NDT) inspection of each parts internal structure. Engine components must pass the NDT inspection before dynamic testing. There are two NDT methods: digital radiography testing (RT) [7] and computed tomography testing (CT) [8]. The deep learning model relies on the generative adversarial model to improve the existing techniques of X-ray image inspection.

The aeronautics engine maintenance crack check processes use X-ray image NDT to find defects in internal hidden areas. The application of deep learning for defect location technology can effectively identify the presence and location of up to eight types of defects, leading to enhanced work quality and efficiency. The finer object detection of image feature maps will lead to more accurate identification of weld flaws than can be accomplished by the standard visual examination. The proposed approach adopts a region-based convolutional neural network and a deep learning neural network for object detection to render an efficient X-ray image diagnosis system. The approach may benefit the inspection work in the aviation industry via increased accuracy and efficiency.

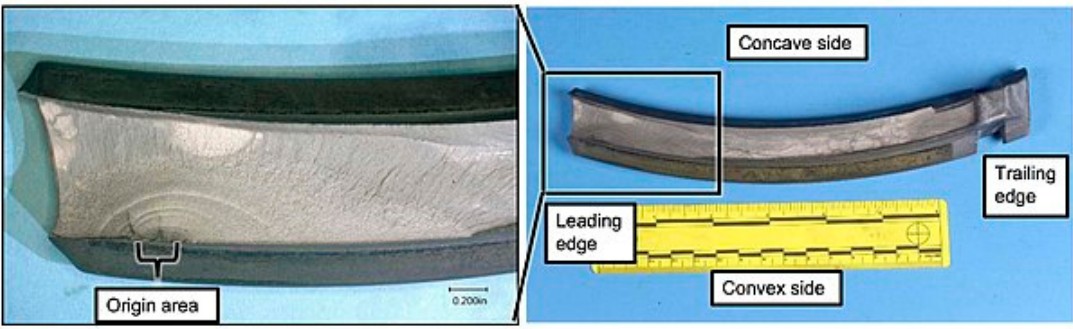

**Figure 1.** Sample of a fracture surface with fatigue [1].

We propose a method of improved transfer learning based on Fast R-CNN that detects composite material hidden cracks from single and multiple NDT radiational images with high reliability. Our neural network, called AE-RTISNet, is based on Fast R-CNN encoder–decoder neural blocks based on residual layers. We produced experimental results for the proposed method. Based on an improved Fast R-CNN, this AE-RTISNet is unique with a simple code and quick to detect the aerial material defect. This region is highlighted with a red square for crack and burn, and manual labeled damage is marked with solid red lines, for the position, size, shape, and direction. Our automatic inspection system of defects discriminates the X-ray image multiple cracks as shown as in Figure 2, and labels the crack between engine blades, as well as differentiates eight types of cracks.

In addition to eight types of defect classification, this experiment also explores the applicability of mutations of Fast R-CNN detection models—for example, the AE-RTISNet improved transfer learning Fast Region-based Convolutional Neural Networks (Fast R-CNN) for a defect detection net. The Fast R-CNN approach requires large amounts of X-ray defect imagery data. Those data facilitate crack image segmentation and the detection of multiple defects of aeronautics engines by automatic edge systems with detection feature extraction and classification processes. Although it is possible to apply deep learning algorithms on datasets, it can be optimized in the explicit domain.

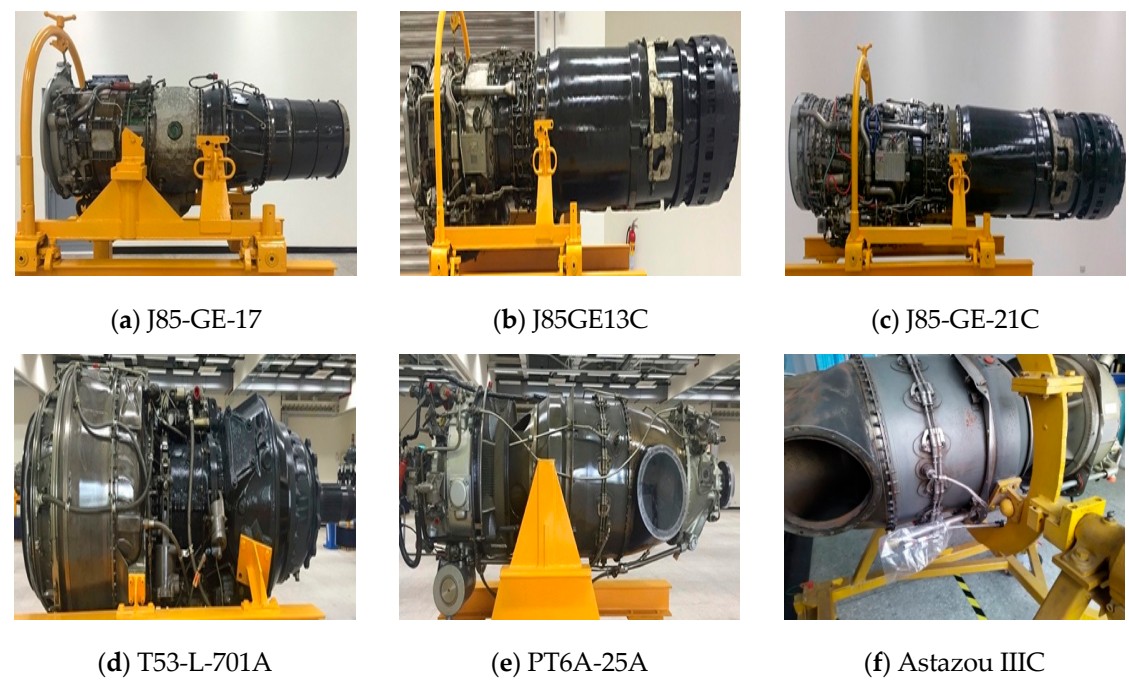

(**a**) J85-GE-17　　　　　　　(**b**) J85GE13C　　　　　　　(**c**) J85-GE-21C

(**d**) T53-L-701A　　　　　　　(**e**) PT6A-25A　　　　　　　(**f**) Astazou IIIC

**Figure 2.** Factory of various military aeronautics engines.

This result empirically shows that a deep learning net complex with pre-tuned model features also yielded superior performance to human crafted features on object identification tasks. We primarily trained for aeronautics engine defects of X-ray image classification for the eight types where sufficient training data existed. Although operators have superior experience on the RT method, we know that radiation is harmful to the human body. The X-ray radiation can take images of objects in real-time [9]. The latter uses AE-RTISNet, tackling both single and multiple detection for X-ray photos of the AI for inspection, and will be the focus of the paper.

The original check work shows the high accuracy of sample type defect detection, as we know that the inspection jobs were all manual. In the inspection process, a technician visually inspects X-ray images of the structures and parts to detect structural cracks and defects. It is easy to have shadow overlap when the sharpness of the X-ray image is poor. The task of the technician in visual inspection is, thus, highly demanding, as she or he may also be subject to fatigue and psychological conditions, and the final diagnostic RT results are determined subjectively by the operators. Thus, the final results are affected by the experience and fatigue and psychological conditions. This detection process is also time consuming and inefficient.

For a resolution to the problem, the aviation industry, including companies, such as Pratt & Whitney, GE, and Honeywell, attempted to automate the inspection process to reduce the burden of the human inspector and assure the quality. An automatic inspection system can assist the inspector to identify the features and defects in X-ray images by marking the object localization window and bounding box automatically. The engine must not have any damage or cracks when in thrusting operation at high speed. It is, thus, of importance to perform the inspection with assured quality and with efficiency. In our paper, the artificial intelligence (AI)-based image processing technique is combined with the NDT inspection technology to improve the aero engine inspection task.

The three main aspects of this are: (a) AI-based function for the training and validation, (b) X-ray images of eight different defects, and (c) the edge-computer based inference system in the engine repair plant. The paper presents the application of deep learning neural network methods in an aeronautics engine X-ray image diagnosis.

The main contributions of the paper are the following:

- A framework of the AI-based NDT inspection for engine parts was developed using fast region-based convolutional neural networks (Fast R-CNN), AE-RTISNet for defect feature detection and description of a multi-task loss function to localize objects for the training and validation.
- The system was trained by collecting a set of X-ray images of engine parts. Eight different defects were labeled in the images for training and validation.
- The proposed system was implemented using an edge-computer-based inference system in an engine repair plant. The effectiveness of the proposed detection in augmenting the inspection capability was demonstrated.

This paper is divided into five sections. In Section 2, the main approach is described. The image data sets, and eight types of defect detection are discussed. The adopted fast region-based convolutional neural network models are then delineated and modifications to the existing network are highlighted. In Section 3, the experiment setup is discussed. In Section 4, the experimental results are discussed. The experimental results indicated that the proposed AI-based inspection method effectively augmented the overall inspection capability. Finally, our conclusions are given in Section 5.

## 2. Main Approach

We used image processing methods for Computer Aided Screening (CAS) [10] of the mark defect detection regions on X-ray images. This section details the image enhancement [11] method that performs an automated multiple defect detection model to apply the maintenance detection Fast R-CNN algorithm to solve the original low pixel X-ray film identification problem. Our work focused on image enhancement, feature segmentation [12], object classification [13], and multiple detections [14]. Our focus was based on addressing the different types of aeronautic engine material hidden crack object classification and detection tasks presented in the following sections.

### 2.1. Image Data Sets Preparation

We collected the main data sets of aerial engine X-ray images from an engine repair plant in Taiwan. The plant is specialized in repairing various types of aerial engines, such as helicopter engines, transport aeronautics engines, and jet engines, etc., As seen from the figures, we had each kind of aerial engine maintenance record for the experiments, as shown in Figure 2.

Figure 3 illustrates the RESCO-MF4 X-ray machine that was used in the experiment equipment. The aerial engine parts are made of metal composite materials. We researched an optimal intelligent transfer learning method with a pre-trained neural net, named AE-RTISNet, with diagnostic results based on the radiographic images with labeled defect regions. For example, if there are 20 engines to be inspected in one month and each engine contains 300 parts, a total of 6000 X-ray images can be collected in one month. If these images are to be inspected by human inspector, the inspection work is very demanding. Hence, automatic inspection by augmenting AI techniques may potentially reduce the workload of human inspectors and assure a more consistent quality.

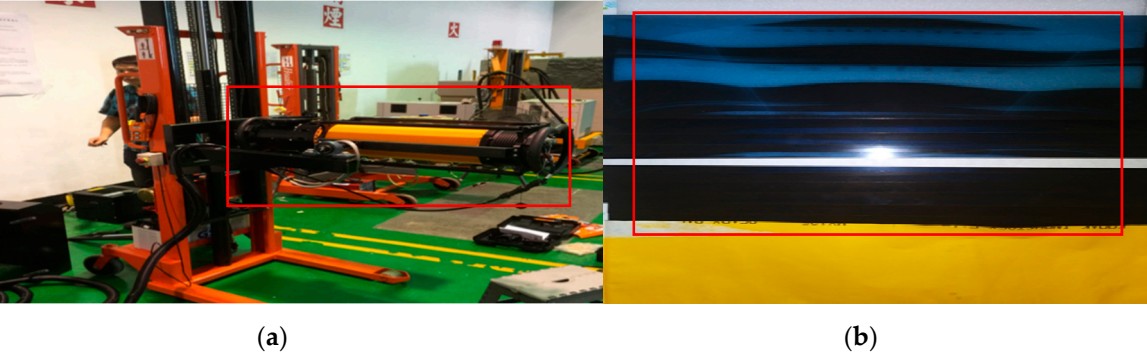

(**a**)                                        (**b**)

**Figure 3.** (**a**) RESCO-MF4 X-ray machine in factory, (**b**) X-ray file.

## 2.2. Eight Type Defects Detection

The image data sets were taken from the archives of the engine repair records. Next, preprocessing using the Python toolkit's image processing library, Python Imaging Library (PIL), was executed to convert the image file format and adjust the output size to a universal lightning memory-mapped database (LMDB) data format [15]. All training and testing tasks were conducted using image data in the LMDB file format. The engine repair plant had accumulated a data gallery of X-ray images.

In the dataset, welds for ACM defects in X-ray images can be categorized as either (a) cracks, (b) incomplete fusion, (c) incomplete penetration, (d) porosity, (e) slag inclusion, (f) undercut, (g) welding spatter, or (h) blowholes as depicted in Figure 4. The X-ray images in the dataset were converted into LMDB format and stored for deep learning applications. In the dataset, certain X-ray images in different light conditions and resolutions were labeled with the above defects. An effort was made to prepare the labeled data in terms of AE-RTISNet. In this endeavor, the eight types of label classes were stored in the first row of the category table string of over 6000 labeled X-ray images from the dataset.

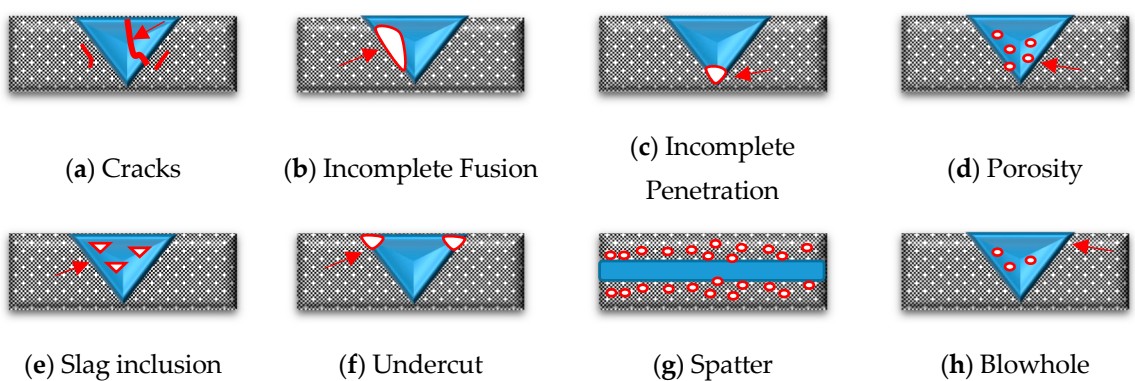

(**a**) Cracks     (**b**) Incomplete Fusion     (**c**) Incomplete Penetration     (**d**) Porosity

(**e**) Slag inclusion     (**f**) Undercut     (**g**) Spatter     (**h**) Blowhole

**Figure 4.** Detection of eight types of defects.

## 2.3. Detection on Description

The AE-RTISNet inspection of X-ray images was required to pre-train the detection of defect types and provide the location or region of the defects. The AE-RTISNet algorithm improved the accuracy used in the X-ray image defect recognition non-destructive radiation testing at the aerial engine maintenance factory. AE-RTISNet was based on the Fast R-CNN algorithm model and developed for detection, localization, and classification. Many different deep neural network models, including DetectNet [16], Fast R-CNN [17], and YOLO [18], have been proposed for object detection and semantic segmentation. In this paper, a model based on the aeronautics engine radiographic testing inspection system was used to determine the bounding box of object defect inspection in X-ray images.

The prepared transfer learning model architecture was evolved from the Fast R-CNN model and Fully Convolutional Network (FCN) frameworks for object detection and semantic segmentation. The layers of Fast R-CNN learn the attention mechanism object detection with a Region Proposal Network (RPN) [19]. The model of Fast R-CNN is, thus, sufficiently flexible and robust to be applicable. The Fast R-CNN model was used to provide the bounding-box object detection for the design of convolutional networks. The normal Fast R-CNN was extended to allow marking of the defect feature maps using pooling layers, which could lead to faster convolution speeds and better accuracy [20].

Under the Jetson™ TX2 embedded computer, the AE-RTISNet compiler for C++ code was used to model the eight types of object classification feature maps and a normal regression was used to estimate the object bounding boxes compiler for the deep learning neural network for object detection [16]. This is an extension to the AE-RTISNet to increase the number of hidden layers. Some changes to accommodate multi-class object detection were also made. This is different from the common practice when applying FCNs [21]. The determination of the structure model for multi-class object detection

contains training and validation steps. Figure 5 illustrates the training and validation procedures in which the L1/L2 loss function is adopted so that the results are not biased to a single class.

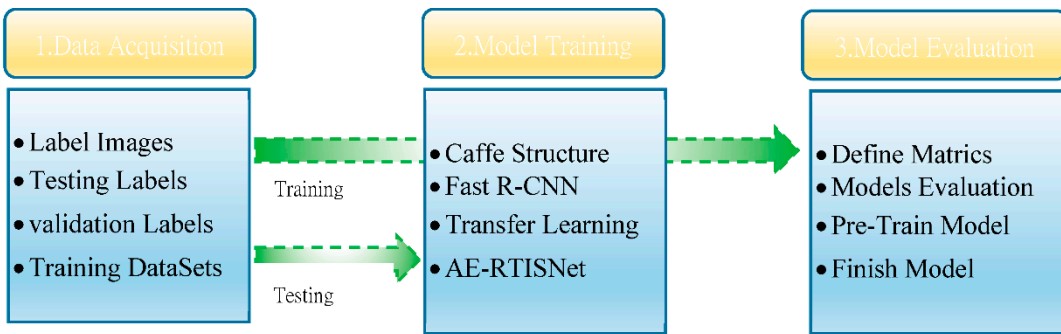

**Figure 5.** Overall architecture function charts.

This paper proposes a deep Fast R-CNN based framework function, which can not only efficiently detect defect damage from X-ray image files but can also be easily adjusted to other aerial part material damage detection problems. We used the Fast R-CNN algorithm model by applying a deep learning algorithm structure, where pre-training the AE-RTISNet model provided the ability to output both the defect regions and the eight types of damages from the input X-ray image datasets. Additionally, the transfer learning method was introduced to reduce the required amount of data and yield high accuracy.

Recently, certain well-known SCI papers demonstrated neural nets that modelled their design on deep learning-based object damage recognition methods. For example, researchers proposed deep layer R-CNN to detect different kinds of cracks while evaluating more than five R-CNN model architectures for the special detection of material corrosion from the input images [22,23]. The Fast R-CNN training algorithm made-up for the disadvantages of R-CNN and SPPnet [17] while improving on their speed and accuracy.

Thus, the Fast R-CNN algorithm method was shown to work with large input datasets that are first cropped into small images of fixed size, and with CNNs applied to classify whether cracks or corrosion are contained in each small image with a fixed size. Thus, the AE-RTISNet model detection gained a more perfect mAP than the R-CNN and SPPnet model. Using the AE-RTISNet model accelerated Fast R-CNN by 10× at test time. The AE-RTISNet model training time was also reduced by 3× due to the faster proposal feature extraction.

Here, we introduce the Fast R-CNN architecture, an input X-ray image, and the defect regions of interest (RoIs) that are input into the fully convolutional network. As in the Figure 6 chart, the ROI is placed into a region fixed-size feature map and quickly mapped to a feature vector by fully connected layers (FCs). We pretrained the transfer learning model with AE-RTISNet, with five max pooling layers and between five and twenty-two conv-layers. We trained the AE-RTISNet network weights with back-propagation of the Fast R-CNN. Then, the AE-RTISNet network is able to update the weights below the pooling layer. AE-RTISNet is proposed as an efficient training method that takes advantage of the feature map sharing during training.

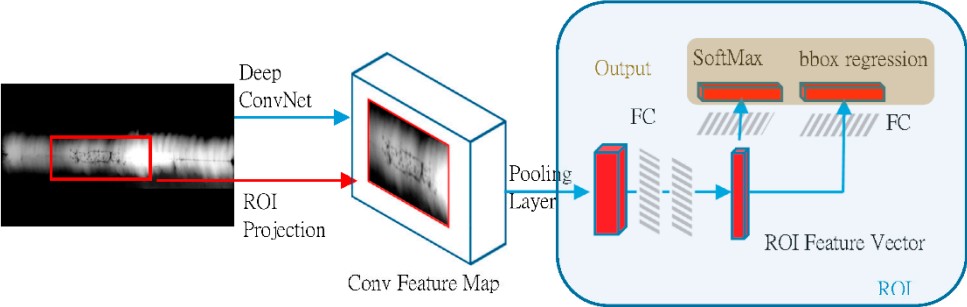

**Figure 6.** Conv Feature Map.

The AE-RTISNet model framework was further developed to enhance the recognition of multiple objects. As a result, unbiased classification for the eight types of object classification feature maps can be obtained. The customized network, as depicted in Figure 7, employs layers to realize an LRN (local area response normalization) [24] and uses a local neural net to establish a set of active competition mechanisms in an attempt to suppress insignificant small feedback neurons for the improvement of the learning performance. Based on a Fast R-CNN model structure, AE-RTISNet has sibling output layers. The AE-RTISNet first processes the whole X-ray image with several convolutional and max pooling layers to produce a convolutional feature map.

$$smooth_{L1}\ (x)\ =\ \begin{cases} 0.5x^2 & if|x| < 1 \\ |x| - 0.5 & otherwise, \end{cases} \tag{1}$$

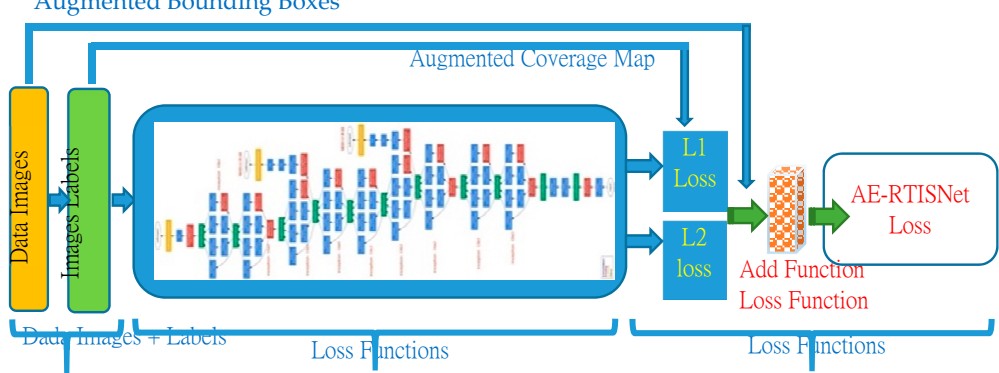

AE-RTISNet Training

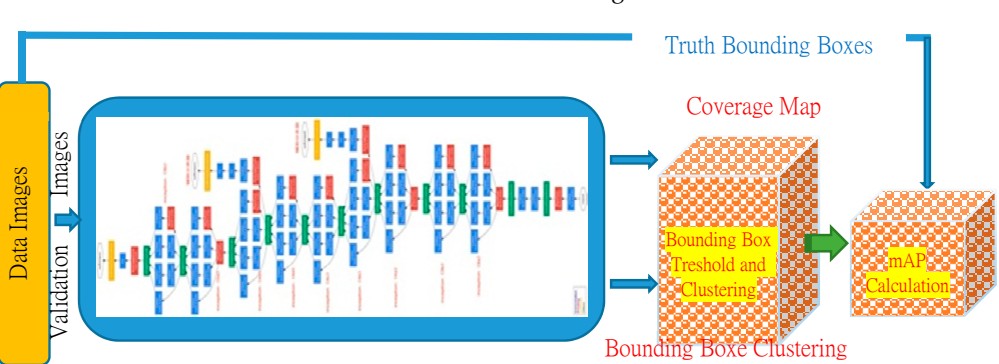

AE-RTISNet Validation

**Figure 7.** $L_1/L_2$ Loss function Validation.

Thus, for each object proposal, a RoI pooling layer extracted a fixed feature vector from the feature map. Each sequence was comprised of fully connected layers, named FC layers. Finally, each set of encoders refined the bounding-box positions for the $K$ classes. The trained RoI was labeled with bounding-box regression offsets, which referred to the $K$ object classes. According to the Fast R-CNN architecture defined algorithm, we know that each RoI training model was labeled with $L_{cls}\ (p,u) = -\log p_u$, which is a log loss, and the $L_{loc}\ (t^u, v)\ =\ \sum_{i \in \{x,y,w,h\}} smooth_{L_1}\left(t_i^u - v_i\right)$, which is a task loss, which together define the $L_{cls}$ and $L_{loc}$ value [25]. For the background Fast R-CNN RoI algorithm, the $L_{cls}$ and $L_{loc}$ are ignored. For the Fast R-CNN model structure of AE-RTISNet on bounding-box regression, we use the loss as the robust $L_1$ loss and $L_2$ loss. These require careful tuning of the learning rates to prevent exploding gradients on Equation (1) [19] and to eliminate this sensitivity.

### 2.4. Inference

The AI-based inspection system was designed to augment the capability of a human inspector to speed up the inspection task and to reduce the likelihood of errors. The Fast R-CNN and AE-RTISNet after training and validation were thus implemented in an embedded processor to perform the defect detection and image segmentation of X-ray images. The embedded platform that was adopted was the NVIDIA® Jetson™ TX2 hardware, which was responsible to perform the on-line inference for defect identification. The inference system was used to upgrade the factory NDT machine equipment, to help the engineer perform the inspection of all X-ray images more efficiently.

## 3. Experimental Setup

### 3.1. System Setup

To develop and verify the proposed AI-based inspection system, a hardware and software environment was required to be setup to support the processing of a large number of data sets for training and inference [26]. The host computer was equipped with an 8-core Intel i7-7700K CPU and NVIDIA-GTX 1660Ti GPU core graphics card. The host computer was responsible for the training of the neural networks. The host computer connected the external expansion device, NVIDIA® Jetson™ TX2 embedded development board, to perform the inference task. The host computer was responsible for the overall data processing and training tasks. Once a model was trained and verified in the host computer, it was compiled and ported to the Jetson™ TX2 embedded development board for inference.

The main operation software was Ubuntu 16.04 version, together with additional installations of NumPy software for C++ and Python programming language interface and the open source software DIGIS (Deep Learning GPU Training System) under the Caffe Deep Learning Framework to provide multi-class training and testing. In the iterations of training, the optimal one-time iteration convergence curve in the trained model was selected as the data for the convergence model. The stored model data were converted to the snapshot image file format by the DIGIS software.

The snapshot model for deployment was then ported to the TX2 to perform the inference task through the Tensor RT software. The deep learning model performed the object identification module for images in the LMDB format, and the self-labeled defect features model was trained to identify the object localization window and bounding box. The input data library data set used the previous layer, the convolutions layer, the max pooling layer, the normalized exponent function SoftMax layer, a 22-layer neural network, and the optimizer calculation model of a stacked deep neural network module. The custom DNN model recognized eight styles and successfully marked the object feature map segmentation.

### 3.2. Network Training

In the experiment, a modified Fast R-CNN was developed and trained to identify the feature map of defect object detection and the segmentation of X-ray images, and the results serve as inputs to the AE-RTISNet in which the characteristic values of the crack defects in the picture were compiled.

The data set was compiled using python code to read the data file format with the extension file name in xml format and input into the neural network layer of the custom DNNs model. The DNNs model was the basic prototype of the neural network model and was made by installing the Caffe Deep Learning Framework [27]. The weight value of the neural network of each layer of the Fast R-CNN on DNNs was adjusted using the python code program compilation.

Some snapshots of the X-ray image data set are depicted in Figures 8 and 9, and illustrate the description screen of the image models. In the presence of defects, a category labeling bounding box is produced, and the regions within the region proposals box is used in the GoogLeNet architecture to perform feature value extraction operations on the trained DNNs. Then, the optimization algorithm Adam is used to refine the model. The process can efficiently distinguish whether it is a defect or a background pixel [28] to reduce the error rate and correct the position of the bounding box through a linear regression model. In addition, the labeling tool ImageImg software is used for the object labeling and compilation as shown in Figure 10. The LabelImg software will automatically check whether there is a corresponding xml file extension tag file in the system so that the overall training can be conducted more efficiently.

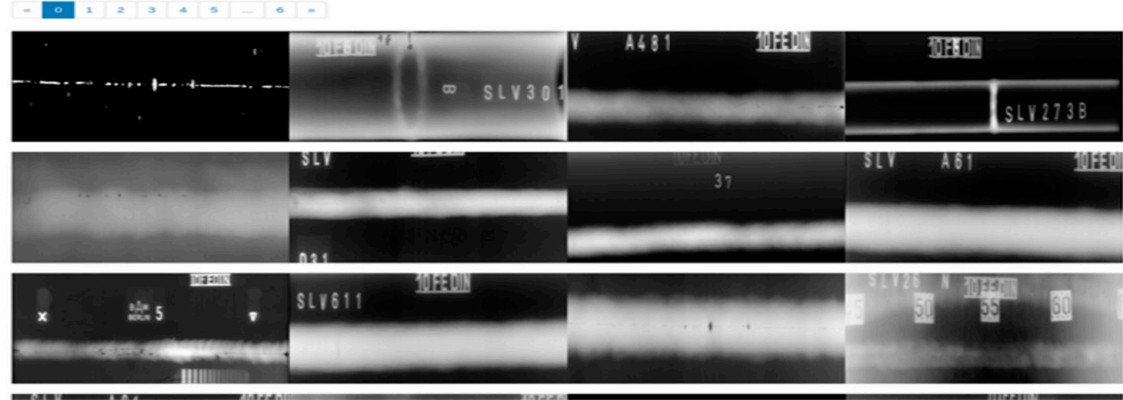

**Figure 8.** X-ray images.

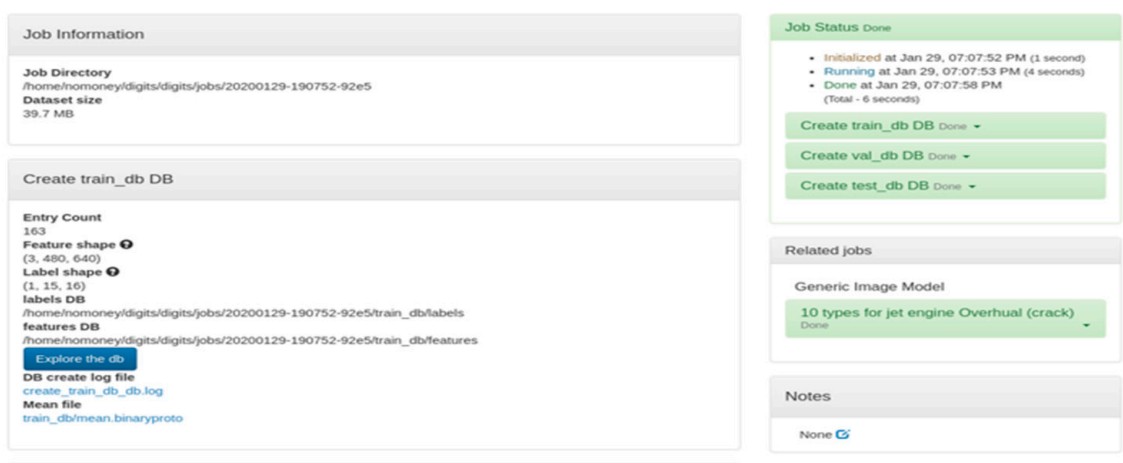

**Figure 9.** Description data in the training process.

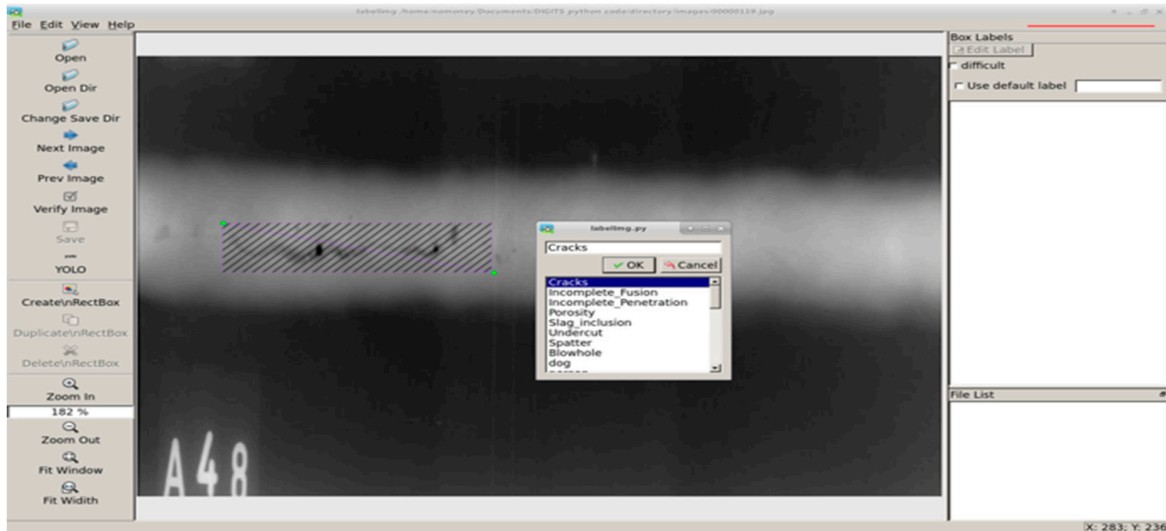

**Figure 10.** Image file labeling.

Both the training and testing were conducted in the determination of the neural network model. To this end, the image file data sets were divided into two folders, among which 66% of the images were used for training, and 33% were used for testing. The experiment used deep networks as a model to identify the X-ray image defect object localization window and bounding box. In addition, AE-RTISNet was used to reduce the error value of the identification trap feature value. The operation follows the description in Figure 9 and, typically, the model was trained for more than 600 iterations. The customized neural network structure plays a role in screening the data. When passing the first layer of convolution, the data are forced to pass a customized layer. The customized layer sharpens the image features so that the features become prominent. As a result, the error rate of the model identification is reduced by 1–2%.

## 4. Experiment Results

Cracks, incomplete fusion, incomplete penetration, porosity, slag inclusion, undercut, spatter, and blowhole are the eight common types of defect characteristics. The maintenance staff according to the engine maintenance manual, must have a NDT Inspector Engineer with FAA, CAA, and EASR certifications to complete the maintenance for aeronautics engines [29]. Our model learns to identify defect categories through image recognition. After the model is compiled, the program is lightweight and can be installed in the Jetson™ TX2 embedded computer. The computer can automatically detect the characteristic values of structural defects, such as engine components, weld beads, and blade bodies.

The test process can identify the type of defect even if the original input X-ray image does not require any preliminary training. The customized DNN and the overall neural network model is evaluated in terms of the mAP (mean average precision). The results confirmed that the model reached the technical standards of a qualified human inspector. The average accuracy (mAP) obtained by the experimental test was 0.82 if the image did not contain a defect and was 0.8 if the picture had certain defects. Figure 11 illustrates an inspection result of an X-ray image cracks and blowhole, etc., and Figure 12 depicts the training performance.

X-ray photograph image checks are used because it is difficult to find a slight line with hidden cracks that have the same length and narrow shapes. This is different from other common wide cracks. We consider that the AE-RTISNet provided the best image region results when compared with the more traditional multiple Fast R-CNN approach.

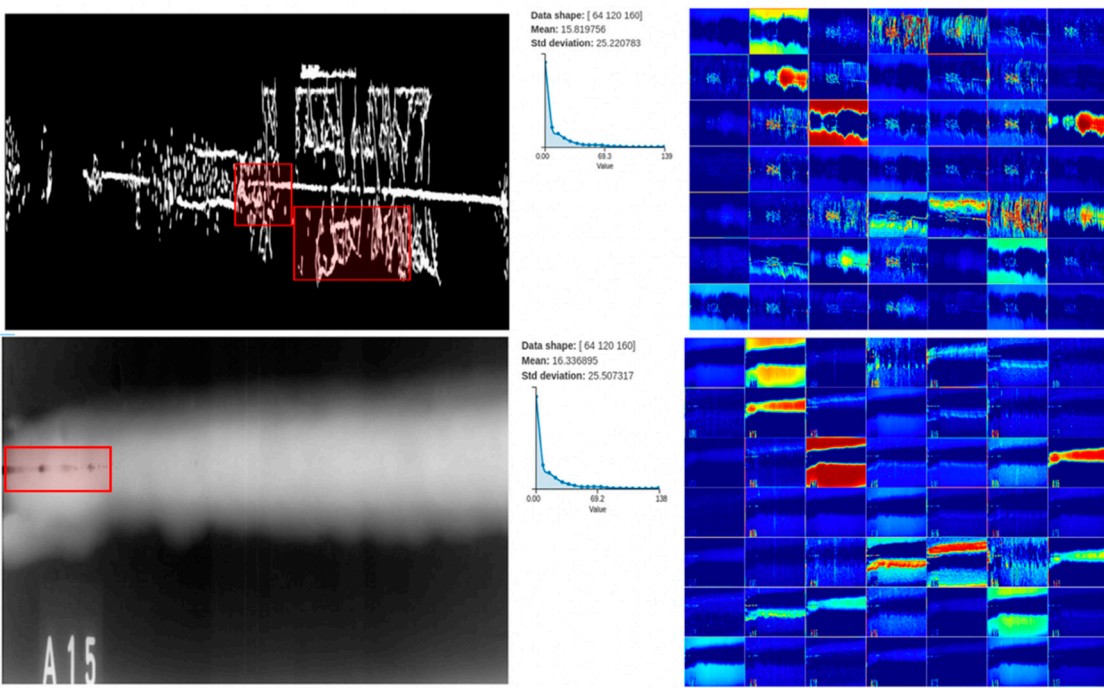

**Figure 11.** Inspection result of an X-ray image cracks and blowhole etc.

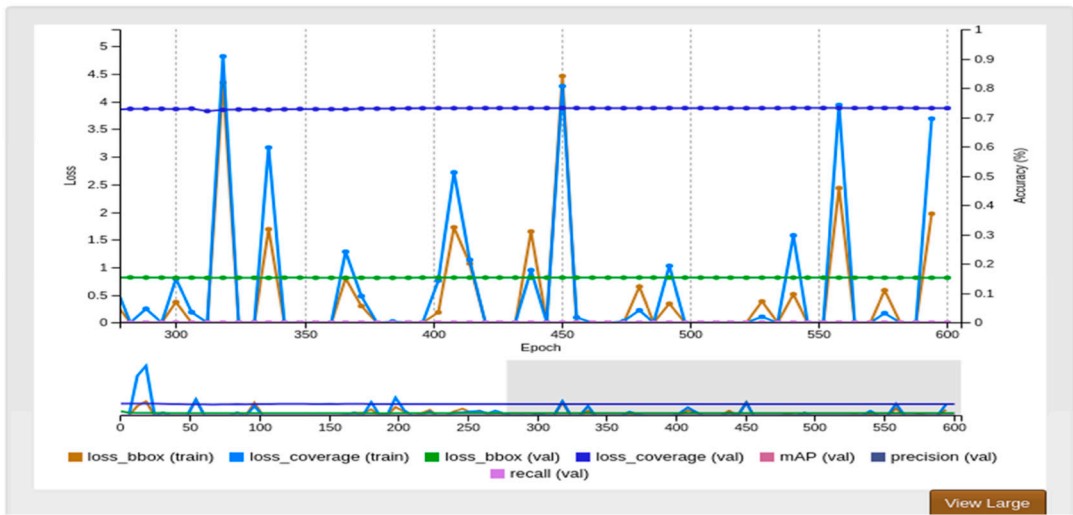

**Figure 12.** Visualization of the training process.

To overcome this issue, we employed a deep learning paradigm of transfer learning tackling both single and multiple detection. We also compared with the original Mask-RCNN, YOLO-V2, and YOLO-V3. Although the test results are listed in Table 1, the accuracy rate of YOLO was slightly higher than that of Fast RCNN, and the Loss score of Fast RCNN was higher than that of YOLO. However, the average elapsed times of the one-stage methods were much less than the two-stage methods.

A field comparison of human visual inspection and the proposed AI-based inspection is shown in Table 2. The experiment verified that the augmentation of an AI-based inspection system for engine diagnosis is feasible and can be an effective tool in quality assurance. The results imply that the maintenance cycle can be made shorter. A preliminary estimate of the reduction in working hours for each engine is more than 60 h, which is equivalent to the savings of a minimum of USD 3 million.



**Table 1.** Inspection results of four methods.

| Method | Accuracy (mAP) | Loss (%) | T(S) |
|---|---|---|---|
| Fast-RCNN | 0.78229 | 0.00238 | 0.2708 |
| Original Mask-RCNN | 0.79129 | 0.00238 | 0.3982 |
| YOLO-V2 | 0.79229 | 0.0238 | 0.0298 |
| YOLO-V3 | 0.72239 | 0.0238 | 0.0326 |

**Table 2.** Manual vs. AI-based inspections.

| Method | Manual Inspection | AI-Based Inspection |
|---|---|---|
| Per 10 Min | 10 | 20 |
| Miss | 1 | 0 |
| Save $ Million Dollars | 0 | 3 |

## 5. Conclusions

Considerable effort has been made so that the approach may benefit in the aviation industry via increased jet engine fans non-destructive testing accuracy and efficiency using AI technology. This paper aimed toward a new method for the AE-RTISNet inspection of the effect of aeronautics engine radiographic testing inspection system to detect defect damage from X-ray image files. This is partly a resulting use of the AE-RTISNet model based on powerful AI technology to prevent non-destructive testing human mistakes in the future. Aiming to clarify the primary issues on AE-RTISNet model testing, this paper provides the relative experimental results of four famous inspection methods, and demonstrates that the accuracy rate of AE-RTISNet was slightly higher than all methods, and had a slightly higher score than the YOLO model. Moreover, the primary types of AE-RTISNet testing and training validation approaches are analyzed and trialled in the paper. Furthermore, case studies on eight types of material defect classifier applications are performed to indicate the feasibility and effectiveness of the proposed quality validation approach. The results achieved a 0.9 mean average precision (mAP) on eight types of material defect classifier problems and required approximately 100 microseconds to develop a deep learning-based inspection procedure to augment non-destructive inspection work in an engine repair plant. Fortunately, the contribution can be achieved by replacing the inherent method with an AE-RTISNet model based on AI to achieve a certain benefit on NDT check time efficiency and save the consumption of labor resources. The experimental results used deep learning technology to determine the parameters of a customized deep learning neural network model to improve the quality assurance in NDT defect detection. The resulting AE-RTISNet successfully deployed C++ code in the TX2 development board. The model was tested and verified as being able to assist the inspector in performing the diagnosis task of the X-ray images in an aeronautics engine repair plant. In particular, the results indicated that the AE-RTISNet-based inspection system can augment the overall inspection system in relaxing the human workload and assuring quality.

**Author Contributions:** Z.-H.C., performed the AE-RTISNet design, have written C++ code and installed in the Jetson™ TX2 embedded computer and wrote the paper. J.-C.J. supervised the project. All authors have read and agreed to the published version of the manuscript.

**Funding:** This research received no external funding.

**Conflicts of Interest:** The authors declare no conflict of interest. The funders had no role in the design of the study; in the collection, analyses, or interpretation of data; in the writing of the manuscript, or in the decision to publish the results.

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
