# Peer review of "AE-RTISNet: Aeronautics Engine Radiographic Testing Inspection System Net with an Improved Fast Region-Based Convolutional Neural Network Framework"

_applsci, doi:10.3390/app10238718_

Round 1

Reviewer 1 Report

The paper looks good although I do not have good enough knowledge about deep learning and AI technology since the content of the paper would be useful to apply the deep learning and AI technology to NDT fields.

My comments are as follows,

  • Please define  AE-RTISNet and DNN.
  • The lower part of AE- RTISNet Validation of Fig. 7 is disappeared
  • Fig. 11 is the result with your method, which is interested to reader. Therefore, please indicate and explain the recognition results indicating the categorized results between (a) cracks –(h) blowhole. In addition, if possible, please indicate more inspection results for each defect type.
  • Please explain about Fig.12 in detail

Author Response

I would like to thank reviewer comment. Under the Fig.12 shows the training results for an inspection result set using the AE-RTISNet network configuration. During training visualization of the training process plots the accuracy and loss values in the top chart. This is handy because it provides real-time visualization into how well or poorly the network is learning. If the accuracy is not increasing or is not as expected you can abort training or delete it using the net model. The learning rate as a function of the training epoch is plotted in the plot.

Reviewer 2 Report

The authors proposed a Fast Region-based Convolutional Neural Networks Framework to detect multiple defect paradigms on X-ray images. The paper is overall interesting and suggested for publication after some modifications. I have a few suggestions that can improve the paper:

  1. The quality f figures should be improved. (figures 1 and 11)
  2. It is suggested to compare you proposed approach with existing models, and present the results in a table with multiple classification metrics.
  3. It is suggested to talk more about the size of the data set, and distribution of those 8 damage classes (if there’s an imbalanced class). Other metrics than accuracy can also be used to evaluate the performance, and more the suitable for evaluating the performance of your multi-class categorization/detection etc.

  1. The contribution of the paper is not clear, it’s suggested to have a separate section mainly talk about the main contribution and scope of the work compared to existing defect detection/classification techniques.

Author Response

I would like to thank reviewer comment. The following is my reply:

  1. We were replaced great quality Fig.1 and Fig.11. Under the Fig.11 shows the training results for an inspection result set using the AE-RTISNet network configuration. During training visualization of the training process plots the accuracy and loss values in the top chart. This is handy because it provides real-time visualization into how well or poorly the network is learning. If the accuracy is not increasing or is not as expected you can abort training or delete it using the net model. The learning rate as a function of the training epoch is plotted in the plot.
  2. It is great reasons for using X-ray photograph image check, because it is hard to find a slight line with hidden cracks have the same length and their shapes are all narrow, which is different from other common wide cracks review. We consider the AE-RTISNet provide more best image region results to the more traditional multiple Fast R-CNN approaches simpler AE-RTISNet, that simple mark regions of interest. To overcome this issue, we employ a deep learning paradigm of transfer learning tackling both single and multiple detection. That is why we were also tried compare with original Mask-RCNN, YOLO-V2 and YOLO-V3. Although, the test results are listed in Table 1, the accuracy rate of YOLO is slightly higher than that of Fast RCNN, the Loss score of Fast RCNN is yet higher than that of YOLO. However, the average elapsed times of one-stage methods are much less than two-stage methods.
  3. In Section 2 we are talk more about the size of the data set, and distribution of those 8 damage classes.
  4. The three main contributions of that are; (a) AI-based function for the training and validation, (b) X-ray images of Eight different defects, and (c) The edge-computer based inference system in the engine repair plant. The paper presents the application of deep learning neural network methods in an aeronautics engine X-ray images diagnosis.

The main contributions of the paper are following:

  • A framework of the AI-based NDT inspection for engine parts is developed by using Fast Region-based Convolutional Neural Networks (Fast R-CNN), AE-RTISNet for defect feature detection and description multi-task loss function to localize objects for the training and validation.
  • The system is trained by collecting a set of X-ray images of engine parts. Eight different defects are labeled in the images for training and validation.

The proposed system is implemented by using an edge-computer based inference system in the engine repair plant. The effectiveness of the proposed detection in augmenting the inspection capability is demonstrated.

Round 2

Reviewer 1 Report

Can I have detailed response against my comments? I need the detailed response against my comments to review manuscript.

Author Response

Point 1: The paper looks good although I do not have good enough knowledge about deep learning and AI technology since the content of the paper would be useful to apply the deep learning and AI technology to NDT fields.

Response 1: Point 1, I would like to thanks reviewer comment. We completely agree with the paper would be useful to apply the deep learning and AI technology to NDT fields. The experiment verifies that the augmentation of AI-based inspection system to engine diagnosis is feasible and can be an effective tool in quality assurance. The results imply that the maintenance cycle can be made shorter. Preliminary estimate of the reduction of working hour for each engine is more than 60 hours which is equivalent to save a lot of million dollars

Point 2: Please define AE-RTISNet and DNN

Response 2: Point2, 

(1) AE-RTISNet: Aeronautics Engine Radiographic Testing Inspection System Net. This is a new DNN layer net data. The AE-RTISNet has improved transfer learning fast region-based convolutional neural networks for defect detection net. AE-RTISNet is unique for simple code and quickly to detection the aerial material defect region is highlighted with mark square for crack region.

(2)DNN: Deep Neural Networks. This is a dis discriminative model that can be trained using back-propagation algorithms. Addition to deep learning algorithm is a branch of machine learning. It is an algorithm that uses artificial neural networks as a framework to represent and learn data

Point 3: The lower part of AE-RTISNet Validation of Fig. 7 is disappeared Fig. 11 is the result with your method, which is interested to reader. Therefor, please indicate  and explain the recognition results indication the categorized results between (a) cracks -(h) blowhole. In addition, if possible, please indicate more inspection results for each defect type. Please explain about Fig. 12 in detail

Response 3: Point.3, 

(1) I would like to thank reviewer command. We have modified the Fig. 7. Under the Fig. 12 shows the training results for an inspection results set using the AE-RTISNet network configuration. During training visualization of the training process plots the accuracy and loss values in the top chart. This is handy because it provides real-time visualization into how well or poorly the network is learning. If the accuracy is not increasing or is not as expected you can abort training or delete it using the net model. The learning rate as a function of the training epoch is plotted in the plot.

(2) The average accuracy (mAP) obtained by the experimental test is 0.82 if the image does not contain a defect and it is 0.8 if the special picture has certain defects. Fog. 11 illustrates an inspection result and Fig .2 depicts the training performance. It is great reasons for using X-ray photograph image check, because it is hard to find a slight line with hidden cracks have the same length and their shapes are all narrow, which is different from other common wide cracks review. We consider the AE-RTISNet provide more best image region results to the more traditional multiple Fast R-CNN approaches simple AE-RTISNet, that simple mark regions of interest. To overcome this issue, we employ a deep learning paradigm of transfer learning tackling both single and multiple detection. That is why we were also tried compare with original Mask-RCNN, YOLO-V2 and YOLO-v3. Although, the test results are listed in Table 1, the accuracy rate of YOLO is slightly higher than that of Fast RCNN, the Loss score of Fast RCNN is yet higher than that of YOLO. However, the average elapsed times of one-stage methods are much less than two-stage methods.

Table1 and Fig. 12 is submitted PDF data.     

Reviewer 2 Report

Authors addressed some of my initial comments. The paper still needs extensive editing of English language and style.

Author Response

Point 1: The authors proposed a Fast Region-based Convolutional Neural Networks Framework to detect multiple defect paradigms on X-ray images. The paper is overall interesting and suggested for publication after some modification. I have a few suggestions that can improve the paper.

  1. The quality figures should be improved (figures 1 and 11)
  2. It is suggested to compare you proposed approach with exiting models and present the results in a table with multiple classification metrics.
  3. It is suggested to talk more about the size of the data set, and distribution of those 8 damage classes (If there's an imbalance class). Other metrics than accuracy can also be used to evaluate the performance, and more suitable for evaluation the performance of your multi-classs categorization/detection etc.
  4. The contribution of the paper is not clear, it's suggested to have a separate section mainly talk about the main contribution and scope of the work compared to exiting defect detection/classification techniques.

Respond 1:point1

I would like to thank reviewer comment. The following is reply:

  1. We were replaced great quality Fig. 11. Under the Fig. 11 shows the training results for an inspection result set using the AE-RTISNet network configuration. During training visualization of the training process plots the accuracy and loss values in the top chart. This is handy because it provides real-time visualization into how well or poorly the network is learning. If the accuracy is not increasing or is not as expected you can abort training or delete it using the net model. The learning rate as a function of the training epoch is plotted in the plot.
  2. It is great reasons for using X-ray photograph image check, because it is hard to find a slight line with hidden cracks have the same length and their shapes are all narrow, which is different from other common wide cracks review. We consider the AE-RTISNet provide more best image region results to the more traditional multiple Fast R-CNN approaches simple AE-RTISNet, that simple mark regions of interest. To overcome this issue, we employ a deep learning paradigm of transfer learning tackling both single and multiple detection. That is why we were also tried compare with original Mask-RCNN, YOLO-v2 and YOLO-v3. Although, the test results are listed Table 1, the accuracy rate of YOLO is slight higher than that of Fast RCNN, the Loss score of Fast RCNN is yet higher than that of YOLO. However, the average elapsed times of one-stage methods are much less than two-stage methods.
  3. In section 2 we are talk more about the size of data set, and distribution of those 8 damage classes.
  4. The three main contributions of that are; (a) AI-based function for the training and validation, (b) X-ray images of English different defects, and (c) The edge-computer based inference system in the engine repair plant. The paper presents the application of deep learning neural network methods in an aeronautics engine X-ray images diagnosis.

The main contribution of the paper are following:

  • A framework of the AI-based NDT inspection for engine parts is developed by using Fast Region-based Convolutional Neural Networks (Fast R-CNN), AE-RTISNet for defect feature detection and description multi-task loss function to localized objects for the training and validation.
  • The system is trained by collection a set of X-ray images of engine parts. Eight different defects are labeled in the image for training and validation.
  • The proposed system is implemented by using an edge-computer based inference system in the engine repair plant. The effectiveness of the proposed detection in augmenting the inspection capability is demonstrated.

Point 2: Authors addressed some of my initial comments. The paper still needs extensive editing of English language and style.

Responds 2: Poin2

I would like to thank reviewer comment. We are looking for MDPI Author Services to request English editing extensive of English language and Style.

Round 3

Reviewer 1 Report

Thank you for your response against my comments. I still have the following comments.

  • The lower part of “AE-RTISNet Validation” of Fig. 7 is disappeared.

  • Fig. 11 is the result with your method, which is interested to reader. Therefore, please indicate  and explain the categorized results between (a) cracks -(h) blowhole by using the recognition results indication  In addition, if possible, please indicate more inspection results for each defect type.

Author Response

Point 1: The paper looks good although I do not have good enough knowledge about deep learning and AI technology since the content of the paper would be useful to apply the deep learning and AI technology to NDT fields.

Response 1: Point1,

Point 1, I would like to thanks reviewer comment. We completely agree with the paper would be useful to apply the deep learning and AI technology to NDT fields. The experiment verifies that the augmentation of AI-based inspection system to engine diagnosis is feasible and can be an effective tool in quality assurance. The results imply that the maintenance cycle can be made shorter. Preliminary estimate of the reduction of working hour for each engine is more than 60 hours which is equivalent to save a lot of million dollars

Point 2: Please define AE-RTISNet and DNN

Response 2: Point2,

(1) AE-RTISNet: Aeronautics Engine Radiographic Testing Inspection System Net. This is a new DNN layer net data. The AE-RTISNet has improved transfer learning fast region-based convolutional neural networks for defect detection net. AE-RTISNet is unique for simple code and quickly to detection the aerial material defect region is highlighted with mark square for crack region.

(2)DNN: Deep Neural Networks. This is a dis discriminative model that can be trained using back-propagation algorithms. Addition to deep learning algorithm is a branch of machine learning. It is an algorithm that uses artificial neural networks as a framework to represent and learn data.

Point 3: The lower part of AE-RTISNet Validation of Fig. 7 is disappeared Fig. 11 is the result with your method, which is interested to reader. Therefor, please indicate and explain the recognition results indication the categorized results between (a) cracks -(h) blowhole. In addition, if possible, please indicate more inspection results for each defect type. Please explain about Fig. 12 in detail

Response 3: Point.3,

(1) I would like to thank reviewer command. We have modified the Fig. 7. Under the Fig. 12 shows the training results for an inspection results set using the AE-RTISNet network configuration. During training visualization of the training process plots the accuracy and loss values in the top chart. This is handy because it provides real-time visualization into how well or poorly the network is learning. If the accuracy is not increasing or is not as expected you can abort training or delete it using the net model. The learning rate as a function of the training epoch is plotted in the plot.

(2) The average accuracy (mAP) obtained by the experimental test is 0.82 if the image does not contain a defect and it is 0.8 if the special picture has certain defects. Fog. 11 illustrates an inspection result and Fig .2 depicts the training performance. It is great reasons for using X-ray photograph image check, because it is hard to find a slight line with hidden cracks have the same length and their shapes are all narrow, which is different from other common wide cracks review. We consider the AE-RTISNet provide more best image region results to the more traditional multiple Fast R-CNN approaches simple AE-RTISNet, that simple mark regions of interest. To overcome this issue, we employ a deep learning paradigm of transfer learning tackling both single and multiple detection. That is why we were also tried compare with original Mask-RCNN, YOLO-V2 and YOLO-v3. Although, the test results are listed in Table 1, the accuracy rate of YOLO is slightly higher than that of Fast RCNN, the Loss score of Fast RCNN is yet higher than that of YOLO. However, the average elapsed times of one-stage methods are much less than two-stage methods.

Point 4: The lower part of “AE-RTISNet Validation” of Fig. 7 is disappeared.

Response 4: Point 4.

(1) I would like to thank reviewer comment. We have modified the Fig. 7. Under the Fig. 12 shows the training results for an inspection result set using the AE-RTISNet network configuration. During training visualization of the training process plots the accuracy and loss values in the top chart. This is handy because it provides real-time visualization into how well or poorly the network is learning. If the accuracy is not increasing or is not as expected you can abort training or delete it using the net model. The learning rate as a function of the training epoch is plotted in the plot.

(2) Cracks, incomplete fusion, incomplete penetration, porosity, slag inclusion, undercut, spatter, and blowhole are the eight common types of defect characteristics. Our model learns to identify defect categories through image recognition after mark red box. The computer can automatically detect the characteristic values of structural defects, such as engine components, weld beads, and blade bodies. Fig. 11 illustrates an inspection result of an X-ray image cracks and blowhole etc., and Fig. 12 depicts the training performance.

Point 5: Fig. 11 is the result with your method, which is interested to reader. Therefore, please indicate  and explain the categorized results between (a) cracks -(h) blowhole by using the recognition results indication. In addition, if possible, please indicate more inspection results for each defect type.

Response 5: Point 5.

I would like to thank reviewer comment. We have modified the Fig. 11. Cracks, incomplete fusion, incomplete penetration, porosity, slag inclusion, undercut, spatter, and blowhole are the eight common types of defect characteristics. Our model learns to identify defect categories through image recognition after mark red box. The computer can automatically detect the characteristic values of structural defects, such as engine components, weld beads, and blade bodies. Fig. 11 illustrates an inspection result of an X-ray image cracks and blowhole etc., and Fig. 12 depicts the training performance. X-ray photograph image checks are used because it is difficult to find a slight line with hidden cracks that have the same length and narrow shapes. This is different from other common wide cracks. We consider that the AE-RTISNet provided the best image region results when compared with the more traditional multiple Fast R-CNN approach.

Table1 and Fig. 7, 11, 12 is submitted PDF data.
